# Effects of Aerated Drip Irrigation on the Soil Nitrogen Distribution, Crop Growth, and Yield of Chili Peppers

**DOI:** 10.3390/plants13050642

**Published:** 2024-02-26

**Authors:** Hongjun Lei, Jinniu Xia, Zheyuan Xiao, Yingying Chen, Cuicui Jin, Hongwei Pan, Zhuodan Pang

**Affiliations:** School of Water Conservancy, North China University of Water Resources and Electric Power, Zhengzhou 450046, China; leihongjun@ncwu.edu.cn (H.L.); xjn1874@gmail.com (J.X.); x201810102054@stu.ncwu.edu.cn (Z.X.); chenyingying@ncwu.edu.cn (Y.C.); b2018081504@stu.ncwu.edu.cn (C.J.); z20211010122@stu.ncwu.edu.cn (Z.P.)

**Keywords:** nitrate-N, ammonium-N, uniformity coefficient, root morphology

## Abstract

In order to study the soil nitrogen (N) distribution pattern in the root zone of chili peppers under aerated drip irrigation (ADI) conditions and analyze the relationship between soil N distribution and crop growth, two irrigation methods (conventional drip irrigation and ADI) and three N levels (0, 140, and 210 kg hm^−2^) were set up in this experiment. Soil samples were collected by the soil auger method at the end of different reproductive periods, and the uniformity coefficient of soil N in the spatial distribution was calculated by the method of Christiansen’s coefficient. The growth status and soil-related indices of pepper were determined at each sampling period, and the relationships between soil N distribution and chili pepper growth were obtained based on principal component analysis (PCA). The results showed that the spatial content of soil nitrate-N (NO_3_^−^-N) fluctuated little during the whole reproductive period of chili peppers under ADI conditions, and the coefficient of uniformity of soil NO_3_^−^-N content distribution increased by 5.29~37.63% compared with that of conventional drip irrigation. The aerated treatment increased the root length and surface area of chili peppers. In addition, the ADI treatments increased the plant height, stem diameter, root vigor, and leaf chlorophyll content to some extent compared with the nonaerated treatment. The results of PCA showed that the yield of chili peppers was positively correlated with the uniformity coefficient of soil NO_3_^−^-N, root vigor, and root length. ADI can significantly improve the distribution uniformity of soil NO_3_^−^-N and enhance the absorption and utilization of N by the root system, which in turn is conducive to the growth of the crop, the formation of yields, and the improvement of fruit quality.

## 1. Introduction

Currently, the overfertilization of nitrogen (N) is widespread in solar greenhouse crop production in China, which leads to an excess of soil N, making it unusable by crops and resulting in a huge risk of N loss from soil. This leads to a decrease in the efficiency of crop water and N use, and environmental pollution (e.g., excessive NO_3_^−^-N content in the groundwater, acidification of the soil, and increased greenhouse gas emissions) [1]. Soil N is an important factor affecting crop growth development and yield [2]. Appropriate irrigation methods can improve the spatial distribution uniformity of soil N and reduce the leaching of N, improving the utilization efficiency of crops [3,4,5,6]. Currently, the most widely used irrigation technology is subsurface drip irrigation (SDI), which can significantly improve water and fertilizer utilization efficiency and achieve precise irrigation and fertilization in the root zone [1,2,3]. However, conventional SDI, in common with other forms of irrigation, is liable to exclude soil air (and therefore oxygen) around the root zone during and following irrigation events [7], resulting in a periodic anaerobic environment in the root zone [8,9], which reduces the soil oxygen content [10] and directly inhibits aerobic respiration of the root system [11]. At the same time, the decrease in root vitality is not conducive to the absorption and utilization of soil water and nutrients by crops [12], which ultimately leads to a decrease in crop yield and fruit quality and becomes the main restriction of high quality and crop yield. Aerated drip irrigation (ADI) technology delivers micro-nano bubble water through the SDI system to the root zone to a certain extent, alleviating the problem of root anoxia caused by conventional SDI. Some studies have shown that aerated irrigation significantly affects the inter-root soil oxygen environment of crops, enhancing soil permeability, improving soil enzyme activity and the activities of soil microorganisms [13,14], promoting the aerobic respiration of the crop root system [15], and increasing crop yields and fruit quality [16]. Soil aeration increases the crop yield and improves the quality of cucumber, potato, zucchini, and rice [11].

The uniformity of soil N distribution remains a key factor in the design and application of different irrigation methods. The uniform distribution of soil N plays an important role in crop growth and yield [17]. Irrigation and fertilization methods will directly affect the distribution of soil N, which in turn influences the nutrient uptake by the root system and ultimately determines the yield and quality characteristics of crops. Previous studies on the relationship between N distribution and crop growth were mostly conducted under conventional drip or sprinkler irrigation conditions [18,19,20], and Zhu et al. [21] showed that the uniformity of soil water and N distribution under sprinkler irrigation conditions significantly affected sugar beet yield. A study by Zhou et al. [20] showed that there was a positive correlation between the uniformity coefficient of soil NO_3_^−^-N distribution and the yield of maize in field drip irrigation trials. The soil NO_3_^−^-N distribution coefficient was positively correlated with corn yield in drip irrigation experiments. Irrigation and fertilization can obviously increase soil water and N content, but the uniformity of N distribution in the soil is still a key factor in the design and application of different irrigation methods. There are fewer studies on the uniformity of soil N distribution under ADI conditions and its effect on crop N uptake and yield and quality. Christensen’s coefficient of uniformity (CU) is often used to assess the degree of uniformity of soil water and N distribution in drip and sprinkler irrigation systems [22] and is able to characterize distribution pattern of the soil water and N. Therefore, in this study, the Christiansen coefficient of uniformity was used to characterize soil N distribution during the reproductive periods with a view to providing a theoretical basis and technical reference for the application and management of water, fertilizer, and soil aeration under ADI conditions.

## 2. Results

### 2.1. Effect of ADI on Physiological Indicators of Chili Pepper Growth

The effect of the ADI on chili pepper plant height and stem diameter is shown in Figure 1. Plant height and stem diameter for different treatments gradually increased after the reproductive periods. During the seedling period, there was no significant difference in plant height and stem diameter in each treatment. In the seedling and fruiting period, plants grew faster, and the plant height and stem diameter of each aerated treatment were higher than those of the nonaerated treatment. As the season progressed, the plant height and stem diameter for the N2A treatment were significantly higher than those of other treatments. During the maturity period, the plant height and stem diameter for the N1A treatment were 112.2 cm and 11.68 mm, respectively, 23.37% and 41.45% (*p* < 0.05) higher than those of the N1C treatment, and the effect of aeration was significant (*p* < 0.05). However, N0A and N2A had no significant difference in plant height and stem diameter compared with N0C and N2C due to non-fertilization or over-fertilization. At the end of the fruiting period, the plant height for the N1A treatment was 97.36 cm, which was 18.65% and 15.13% higher than that of N0A and N2A treatments, respectively. The stem diameter for the N1A treatment was 11.25 mm, which was 12.17% higher than that of the N0A treatment during the same period, and there was no significant difference. The trends of plant height and stem diameter in each treatment were basically consistent with that in the maturity period. The changes in chlorophyll content of chili pepper leaves in different treatments are shown in Figure 2. The chlorophyll content of each treatment during the reproductive period showed a trend of increasing, then decreasing, and finally tended to be stable. The chlorophyll content of each treatment reached the maximum value in the flowering and fruiting period. Compared with the conventional dripping treatment, the chlorophyll content of chili peppers with the aerated treatment increased by 17.62% on average (*p* < 0.05). Compared with the treatment without N, the chlorophyll content in low and normal N treatments increased by 8.84% (*p* < 0.05) and 12.60% (*p* < 0.05), respectively.

### 2.2. Effects of ADI on the Spatial Distribution of Soil NO_3_^−^-N, NH_4_^+^-N Contents

As shown in Figure 3, the average NO_3_^−^-N content of N2C, N1C, and N0C treatments in the maturity period was 136, 124, and 106 mg kg^−1^, respectively (*p* < 0.05). The average NO_3_^−^-N content of N2A, N1A, and N0A treatments in the maturity period was 106, 97, and 83 mg kg^−1^, respectively, indicating a decrease of 11.36%, 27.45%, and 28.58%, respectively. Regardless of whether ADI or conventional drip irrigation was employed, the content of NO_3_^−^-N in the middle layer (10~20 cm) was lower, the content of NO_3_^−^-N in the upper and lower layers (0~10 cm, 20~30 cm) was higher, and the difference in NO_3_^−^-N content in the upper and lower layers for the N2A treatment was the largest. Soil NO_3_^−^-N content increased with the increase in soil depth in conventional SDI treatments. The NO_3_^−^-N content in the lower layer (20~30 cm) was 28.56%, 28.89%, and 47.10% higher than that in the upper layer (0~10 cm) under conventional drip irrigation treatment conditions in the maturity period, respectively (*p* < 0.05), and was higher than that of the ADI treatment in the corresponding period. The NO_3_^−^-N content in the middle soil layer under ADI and conventional SDI conditions was 96 and 78 mg kg^−1^, respectively (*p* < 0.05). After each growth period, the NO_3_^−^-N content of each treatment profile gradually increased, and the average NO_3_^−^-N content of each soil layer in the maturity period was 25.29~90.86% higher than that in the flowering and fruiting period. In the horizontal measurements, the average NO_3_^−^-N content of each soil layer away from the drip head under ADI conditions was 98 mg kg^−1^ and was 67% higher than that of conventional SDI. The difference in NO_3_^−^-N content between ADI and conventional SDI was the most significant during the fruiting period. The distribution of NO_3_^−^-N in the 0~30 cm soil layer under ADI conditions is shown in Figure 4. Overall, with the deepening of the soil layer, the content of NO_3_^−^-N decreased, and the transversal transport rate decreased over time. The soil N content under ADI conditions was significantly lower than that of conventional SDI, and the NO_3_^−^-N content changed from deep to shallow in different treatments. The total NO_3_^−^-N in the 0~30 cm soil layer of chili peppers during the same growing period under ADI conditions was significantly lower than that of the nonaerated treatment. The trend of ammonium-N (NH_4_^+^-N) in different treatments is shown in Figure 5. It revealed that there was little difference in the content of NH_4_^+^-N in the 0~30 cm soil layer in the maturity period among different treatments, the content of NH_4_^+^-N in all soil layers in the maturity period was the lowest in the ADI treatment, and the content of NH_4_^+^-N in the N0A treatment was the lowest. Throughout the soil profile, under conventional SDI conditions, the content of NH_4_^+^-N in the middle soil layer was higher than that in the upper and lower soil layers, which was opposite to the distribution of NO_3_^−^-N. In different fertilization treatments of ADI, the differences in NH_4_^+^-N between different treatments in each soil layer were small, indicating that the NH_4_^+^-N content in each layer was less affected by the aerated treatment.

### 2.3. Effect of Different Treatments on the Coefficient of Uniformity of Distribution of NO_3_^−^-N and NH_4_^+^-N

Soil N distribution was mainly affected by water transport and crop physiological activities, as shown in Table 1. With the development of the reproductive period of chili peppers, the uniform coefficient of NO_3_^−^-N in each soil profile of ADI treatments gradually increased, while the uniform coefficient of NH_4_^+^-N tended to stabilize. The average NO_3_^−^-N distribution coefficient of the soil profile in the N2C treatment was the lowest during the whole growth period. The soil NO_3_^−^-N coefficient of N2A and N0A under aeration drip irrigation conditions was 0.91, and the difference was not significant, while the NO_3_^−^-N coefficient of the N1A treatment was lower. In the conventional SDI fertilization treatments, the soil profile of the N2C treatment had the lowest NO_3_^−^-N distribution coefficient at only 0.48, and the coefficient was lower than 0.6 in all growth periods except for the flowering and fruiting period. The NO_3_^−^-N distribution coefficients of ADI treatments were significantly higher than those of the conventional drip irrigation treatments during all growth periods, and the increase was between 17.33% and 65.45% (*p* < 0.05). There was no significant difference in the distribution coefficient of NO_3_^−^-N among different ADI treatments. NO_3_^−^-N is the main form of N absorbed by the chili pepper. NH_4_^+^-N, as a substrate of the nitrification reaction, directly affects the changes in NO_3_^−^-N content and absorption and utilization of crops. As shown in Table 2, in the 0~30 cm soil profile, there was no significant difference in the NH_4_^+^-N distribution coefficient among different treatments, and the N2C treatment had the lowest NH_4_^+^-N distribution coefficient during the fruiting period. Compared with other treatments, there was no significant difference in other growth periods. For the whole soil profile, under conventional SDI conditions, the content of NH_4_^+^-N in each soil layer was not significantly different, but under ADI conditions, the content of NH_4_^+^-N was higher in the whole growth periods of peppers. The difference in the coefficient of uniformity of NH_4_^+^-N in the soil profile in each treatment was small.

### 2.4. Effect of ADI on Root Morphology and Root Vigor of Chili Peppers

Soil aeration and N application had significant effects on the root length and root surface area of chili plants, and they were positively correlated with the root length and root surface area. Table 3 shows that both the N application and aerated treatments increased the root length and root surface area of chili peppers. ADI increased the root length, root surface area, and root volume by 54%, 42%, and 62%, respectively, compared with the conventional SDI treatment. The root length and root surface area increased by 100.03% and 115% for the N2A treatment, 131% and 137% for the N1A treatment, and 68% and 38% for the N0A treatment (*p* < 0.05) compared with that of the N0C treatment. The root length, root surface area, and root volume of N2A and N1A treatments were reduced by 14.97%, 10.09%, and 19.58%, respectively (*p* < 0.05). Aeration significantly promoted the increase in the root length, root surface area, and root volume of chili peppers. As shown in Figure 6, compared with no N treatment, the root activity of normal and low-N treatments was increased by 21.14% and 22.19% (*p* < 0.05), respectively. Compared with no N treatment, the root activity of ADI was increased by 21.86% and 14.68% (*p* < 0.05). The root vigor of the N2A treatment was the greatest in the flowering and fruiting period and was significantly increased by 123.07% (*p* < 0.05) compared with that of the N0C treatment. The results showed that the root length and root surface area of chili peppers were significantly affected by ADI, and in the fruiting period, ADI improved root activity more than conventional SDI. The positive feedback effect of ADI on root activity of peppers decreased with the extension of the growth period.

### 2.5. Effect of ADI on Yield, Fruit Quality, and N Uptake of Chili Peppers

According to the yield and quality analysis presented in Figure 7, ADI resulted in an average yield increase of 92.5% compared to conventional SDI (*p* < 0.05). Additionally, the normal N treatment exhibited an average yield decrease of 5.54% compared to the low-N treatment and an average yield increase of 53.28% compared to the no N treatment. The yield of the N1A treatment reached the maximum value, i.e., 6.22 t hm^−2^; it was 152.85% higher than that of the N1C treatment, and the aeration effect was significant (*p* < 0.05). The ADI treatment increased the Vc content of chili peppers by an average of 31.15% (*p* < 0.05) compared with the conventional SDI treatment. The N1A treatment increased the Vc content of chili peppers by 51.63% (*p* < 0.05) compared with the N1C treatment, and the N1A treatment increased the Vc content of chili peppers by 9.48% (*p* < 0.05) compared with the N2A treatment. The fruit soluble protein content of the aerated treatment increased by 20.64% compared with that of the nonaerated treatment (*p* < 0.05). The soluble solids content of N2C, N1C, and N0C was 2.96, 2.78, and 2.28 g, respectively. The soluble solids content of aerated drip irrigated fruits was increased by 73.65%, 53.6%, and 61.84% at the same N application rate.

As can be seen from Figure 8, the cumulative N uptake of fruits, leaves, stems, and roots decreased successively, which was in line with the growth law of reproductive growth during the maturity period. The total N accumulation of chili pepper organs among different treatments was N2A > N1A > N2C > N1C > N0A > N0C. The total N accumulation increased by an average of 13.29% in the normal N treatment compared with that of the low-N treatment, and cumulative N uptake increased by 33.39% in the aerated drip treatment compared with that of conventional subsurface drip treatment. This indicated that the increase in N fertilizer application and root aeration could improve the cumulative N uptake of chili peppers to a certain extent. The cumulative N uptake of chili peppers can be increased to some extent by increasing the amount of applied N fertilizer and root aeration. However, the cumulative N uptake of the low-N aerated treatment was 9.97% higher than that of the normal N and nonaerated irrigation treatment, which showed that the cumulative N uptake of chili peppers could be further increased by ADI with a reasonable level of soil N application.

### 2.6. Relationship between the Mineral N Distribution Coefficient of Uniformity in the ADI Soils and Physiological Indices, Root Indices, and Yield of Chili Peppers

Principal component analysis (PCA) was adopted to evaluate the trade-offs and synergistic relationships among 10 indicators of the physiological growth, yield, and spatial distribution uniformity coefficient of soil N of chili peppers for different treatments (Figure 9). The cumulative contribution of the first and second principal components was 66% of the total variance. The treatment cluster analysis revealed that the N0C treatment exhibited the highest degree of similarity among the three replicates, distinguishing it more significantly from the N0A treatment. Additionally, the N1C treatment demonstrated stronger differentiation in principal component 1 than the N1A treatment, whereas the N2C and N2A treatments showed stronger differentiation in principal component 2. The results showed that there was no correlation between yield and uniformity coefficient of soil NH_4_^+^-N and a positive correlation between yield and other indices. The distribution uniformity coefficient of soil NO_3_^−^-N, root activity, and root length had the strongest positive correlation with yield, indicating that ADI could improve the root activity and root length of chili peppers and promote the uniformity of soil NO_3_^−^-N distribution. The distribution uniformity coefficient of soil NH_4_^+^-N was positively correlated with the root surface area and root volume, while the uniformity coefficient of soil NO_3_^−^-N was negatively correlated with the root surface area. There was a significant positive correlation between plant height and stem diameter and a synergistic relationship with the root activity, root length, root surface area, and root volume. It is worth noting that the distribution coefficient of soil NH_4_^+^-N was negatively correlated with that of NO_3_^−^-N, and the contribution of the NO_3_^−^-N distribution coefficient to the principal component was greater than that of the NH_4_^+^-N distribution coefficient, indicating that the soil NO_3_^−^-N coefficient had a more positive effect on the growth and development of chili peppers and yield. The Chlorophyll content of leaves was positively correlated with the yield, root vigor, and root length.

## 3. Discussion

### 3.1. “Response Relationship” among the ADI–Crop Growth–Soil Mineral N Distribution

ADI promotes crop growth, fertilizer uptake, and utilization by altering the soil microenvironment, which in turn affects soil water, fertilizer transport, and transformation [23]. In this study, it was found that soil N distribution was stable throughout the reproductive period of chili peppers under ADI conditions, whereas the change in soil N content fluctuated greatly under conventional SDI treatment conditions. The soil nutrient content varied greatly in the spatial distribution throughout the growth period, which was not favorable for crop growth and development [24,25]. The soil N distribution law of ADI was different from that of conventional SDI, resulting in different root distribution patterns. The high N content of water near the drip irrigation head led to the increase in the root system. In the seedling period and fruiting period of peppers, the root vitality of peppers under ADI conditions was stronger, thus absorbing more water and nutrients, which was conducive to the increase in yield and the improvement of quality. Crop absorption and utilization of soil N is helpful to further improving the spatial distribution uniformity of soil N. In ADI, the soil layer where the irrigator was located was the main root layer of chili peppers. Plant roots in the higher irrigation treatment absorbed more water [26], promoting the uniformity of N distribution to a certain extent.

The distribution pattern of N in the soil profile was influenced by many factors such as soil water movement, soil fertility, and N form [27]. In this study, the soil NO_3_^−^-N content of each ADI treatment was lower than that of the nonaerated treatment. Additionally, the N content in 0~10 cm and 20~30 cm soil layers was higher than that in 10~20 cm soil layers in all treatments. This was because urea was rapidly hydrolyzed to NH_4_^+^-N shortly after its application to the soil and then completely oxidized to NO_3_^−^-N within 1 to 2 weeks under the effect of ammonia-oxidizing microorganisms [28]. During the transformation process, urea was easily volatilized into the air and leached to deeper soil layers [29], resulting in lower N content in the till layer of the soil. Conversely, chili peppers irrigated with aerated drip systems developed more robust root systems due to the increased root vigor, thereby enhancing the uptake and utilization of NO_3_^−^-N in the soil. Under ADI conditions, the coefficient of uniformity of NO_3_^−^-N distribution in the vertical direction of the soil profile significantly surpassed that of nonaerated treatments. This improvement was primarily attributed to the enhancement of the root growth environment and the heightened nutrient requisitioning capabilities of the root system [30]. Furthermore, aeration increased internal pores within the soil, thus enhancing water circulation. Consequently, the transportation of soil NO_3_^−^-N with water was facilitated [31], thereby improving the distribution uniformity of NO_3_^−^-N in the soil. Additionally, the results showed that NH_4_^+^-N content in different soil layers had no significant difference during the reproductive period of chili peppers. The observed fluctuation was very small, indicating that the N content of the 10~20 cm layer was higher than that of the 0~10 cm and 20~30 cm layers. These findings are in line with those reported by Wu et al. [32]. The motivation for this analysis lies in the inherent instability of NH_4_^+^-N in the soil matrix, where NH_4_^+^-N is rapidly converted to NO_3_^−^-N through the nitrification process [33]. The observed peak concentration of NH_4_^+^-N near the droplet can be attributed to the predominantly negative charge of soil colloids [34], which tend to adsorb positively charged NH_4_^+^-N ions immediately adjacent to irrigation water sources.

### 3.2. Effect of Soil Moisture Content and Mineral N Distribution Uniformity Coefficients on Crop Production under ADI Conditions

Soil N content and its distribution uniformity directly affect nutrient uptake, accumulation and distribution of assimilates, and balanced growth above ground and below ground [35]. N is an important element that constitutes the structure of chlorophyll, and leaf chlorophyll content can positively represent the N nutritional status of crops [36]. In this study, at the higher N application rate (210 kg hm^−2^), the chlorophyll content of leaves was higher, but the N accumulation of fruits in N2C and N1C treatments was lower than that in N2A and N1A treatments. Obviously, the soil N condition under conventional SDI treatment conditions was not conducive to the transfer of nutrients from vegetative organs to reproductive organs, resulting in a decrease in yield [37]. This in turn reduced the N fertilizer bias productivity. In addition, the chlorophyll content of leaves treated with N1A and N2A was higher, and the difference was not significant. It was concluded that under ADI treatment conditions, an appropriate N fertilizer application rate and the higher uniformity coefficient of NO_3_^−^-N could promote the nutrient uptake and utilization of crops. Chili pepper yield and cumulative N uptake under ADI conditions showed a consistent pattern. The cumulative N uptake and yield of crops in the N fertilizer (140 kg hm^−2^) treatment were significantly higher than those under conventional SDI conditions, which was in agreement with the findings of Zhong et al. [38]. However, maize experiments showed that there was no correlation between yield and soil moisture and N distribution uniformity when soil moisture and N were sufficient [39], probably due to the different test crops, N fertilizer application rates, and irrigation methods. A higher spatial distribution uniformity of soil N was more favorable to the growth and development of crop roots, while the activities of key enzymes maintaining N metabolism were higher when the soil condition was better aerated [40], further improving N uptake and utilization by chili peppers. Compared with the conventional SDI treatment, the ADI treatment increased N utilization efficiency. But the soil N content of aerated and conventional drip irrigation treatments under normal N conditions was higher, which could lead to ineffective nutrient losses [41]. Therefore, although ADI can improve the nutrient uptake and utilization of chili peppers, the irrational fertilizer application (210 kg hm^−2^) led to excessive soil N content, resulting in a large N leakage to the bottom of the root zone of crops with water infiltration. This increased the risk of deep leaching [42,43]. While the soil N content in N0A and N0C treatments was low, and the root vigor was lower than that of each fertilization treatment, the root system was unable to obtain sufficient nutrients from the soil, thus inhibiting crop growth and yield formation [44,45]. It can be seen that soil N distribution uniformity is not an important guarantee for high yield and N use efficiency. Only when the content of N fertilizer in soil is within the appropriate range can the higher N uniformity coefficient effectively promote the growth of chili peppers and increase the yield.

## 4. Materials and Methods

### 4.1. Overview of the Test Site

The experiment was carried out in an intelligent solar greenhouse of the Experimental Farm of Efficient Water Use in Agriculture of North China University of Water Resources and Hydropower (34°47′5.01″ N, 113°46′54.61″ E). The modern greenhouse has a span of 9.6 m, a total area of 537.6 m^2^, and an opening of 4 m. The experiment was conducted from 16 March to 4 July 2022. The maximum temperature of the greenhouse microclimate was 37.2 °C, the minimum temperature was 10.4℃, the maximum humidity was 62.2%, and the minimum humidity was 34.7%. The test soil type was clay loam with a soil capacity of 1.25 g cm^−3^. The basic soil properties were as follows: an available N mass ratio of 38.87 mg kg^−1^, Olsen-P mass ratio of 9.98 mg kg^−1^, available potassium (K) mass ratio of 3.42 mg kg^−1^, organic matter (OM) mass ratio of 21.54 mg kg^−1^, and soil pH of 7.32.

### 4.2. Experimental Design

Chili pepper variety “Jinfu 807” was used as the test crop, and a completely randomized block design was adopted by setting an ADI system in combination with no N addition (N0A), low N addition (N1A), normal N addition (N2A), or a conventional drip irrigation system in combination with no N addition (N0C), low N addition (N1C), and normal N addition (N2C). The low and normal N addition were 140 kg hm^−2^ and 210 kg hm^−2^. There were three replications for each treatment with 10 chili pepper plants per replication (Table 4). N, phosphorus, and potash fertilizers were used along with urea (N ≥ 46%), calcium superphosphate (P_2_O_5_ ≥ 16%) and potassium sulfate (K_2_O ≥ 51%), respectively. Before planting, 150 kg hm^−2^ of phosphate fertilizer and 200 kg hm^−2^ of potash fertilizer were applied basally, and urea was used for the N fertilizer, which was applied 5 times, following a N topdressing dosage ratio of 2:3:2:3:1. Irrigation was carried out according to the cumulative evapotranspiration (Ep) of a standard 20 cm evaporation tray every 5–7 days after the rest period of the seedlings.

The experimental layout is shown in Figure 10. Each small test plot was 4 m long and 0.6 m wide, totaling 2.4 m^2^. The planting layout was 33 cm per plant and 60 cm apart. A JOHNDEERE drip pipe (John Deere, Inc., Morelin, IL, USA) was used for water supply. The inner diameter of the drip tape was 16 mm, the wall thickness was 0.6 mm, the rated emitter flow rate was 1.2 L h^−1^, the spacing of the drip head was 33 cm, the maximum working pressure was 2.0 atm, and the burial depth of the drip tape was 15 cm. A 60 cm deep plastic film was laid between each small area to prevent the side seepage of water. Both aerated irrigation and conventional irrigation were supplemented by SDI, and the level of irrigation was calculated according to the cumulative value of Φ20 cm of the standard evaporation disc between two irrigation intervals. The water supply pipe of each plot was controlled separately, and an intelligent water meter recorded the amount of water irrigated. The seedlings of peppers of 8~10 leaf age were selected for transplantation. The seedlings were planted on 16 March 2022, and they were watered thoroughly on the same day to ensure the survival of the seedlings. The seedling period was 7~10 days, and mulching was carried out on 28 March (22 days after transplanting).

### 4.3. Sampling and Measurement Methods

#### 4.3.1. Determination of Physiological Indices of Growth in Chili Peppers

Five evenly grown plants were selected for each treatment, and the plant height was measured every 10 days with a tape measure from the root of the pepper to the growing point of the pepper. The stem diameter was measured by the crisscross method using a vernier caliper with an accuracy of 0.01 mm.

The chlorophyll content of chili peppers was measured every 10 d using a chlorophyll meter (SPAD-502, Konica Minolta Corporation, Tokyo, Japan).

#### 4.3.2. Cumulative N Uptake of Chili Pepper

At the end of each reproductive period, five chili pepper plants with uniform growth were harvested. The roots, stems, leaves, and fruits were separated and heated in an oven at 105 °C for 30 min and then dried at 75 °C until the weight was constant. The samples were then ground in a plant pulverizer and then passed through a 0.25 mm sieve. A total of 0.5 g of the sample was placed in a digestion tube, 8 mL of concentrated H_2_SO_4_-H_2_O_2_ was added, and they were digested for 2–3 h at 400 °C. The total N content was determined by a Kjeldahl N analyzer (K9840, Haiergy Future Technology Group Co., Ltd., Jinan, China), and the accumulated N content was calculated based on the corresponding dry mass.

#### 4.3.3. Measurement of Root Indicators

At the end of each reproductive period, the root system was removed as much as possible and was placed in a mesh bag with a diameter of 0.5 mm. After being soaked in water in the laboratory, it was rinsed with water to separate the soil from the roots. After the roots were washed, the surface of the root system was blotted with absorbent paper, and the fresh mass was weighed. The root system was scanned into a grayscale map using a root scanner (Epson Expression 1600pro, Epson Corporation, Beijing, China) and was organized into a transparent tray in order to keep the branches of the root system from intertwining during scanning. The acquired TIF maps were analyzed by the Win RHIZO Pro image processing system to obtain the effective root surface area (cm^2^), effective root volume (cm^3^), and root length (cm).

Using the TTC (triphenylte trazolium chloride) method, 0.5 g of root tip samples was weighed and submerged in a mixture of 5 mL TTC (purity 98.6%, Sigma Reagent Co., Ltd., St. Louis, MO, USA) and phosphate buffer in a 20 mL beaker in a dark incubator. After treatment in the dark at 37 °C for 2 h, 2 mL 1 mol L^−1^ sulfuric acid was added to each beaker to stop the reaction. The root system was rinsed with deionized water once. After draining the water, 5 mL of acetone solution (20%) was added to the beaker to extract the 2,3,5-triphenylformazan in the root sample, and then the extent of TTC reduction was determined using a UV spectrophotometer [46].

#### 4.3.4. Sampling, Determination of Soil NO_3_^−^-N, NH_4_^+^-N Concentration, and the Calculation of Their Coefficient of Uniformity

At the end of each reproductive period of chili peppers, soil samples were taken in layers along an S-shaped route by the soil auger method. Six sampling points were taken for each treatment, with one soil layer for every 10 cm for a total of three soil layers, and fresh soil samples were stored in a refrigerator. Soil samples were extracted with 2 mol L^−1^ potassium chloride solution, and NO_3_^−^-N and NH_4_^+^-N contents were determined using a continuous flow analyzer (AA3 HR Auto Analyzer, Seal Analytical Norderstedt, Germany). The degree of uniformity of soil N distribution was assessed using Christensen’s coefficient of uniformity, and the calculation formula is as follows:(1)$$Cu=1-\frac{\sum_{i=1}^{n}\left|\theta_{i}-\bar{\theta }\right|}{n\bar{\theta }}$$
where *Cu* denotes the coefficient of uniformity of soil N distribution (%), $\bar{\theta }$ denotes the mean value of N content at different soil depths (mg kg^−1^), $\theta_{i}$ denotes the N content in each sampling point (mg kg^−1^), and n denotes the depth number of the sampled soil. The soil sampling depths were 0~10 cm, 10~20 cm, and 20~30 cm.

### 4.4. Statistical Analysis

Data analysis and graphing were performed using Excel 2022 for data processing. The analysis of variance (ANOVA) was conducted using the ANOVA process of SPSS 19.0. The significance of differences between treatments was tested using the Least Significant Difference (LSD) method, correlation analysis was carried out using Pearson’s correlation coefficient, and correlation graphs were plotted using Origin 2021. The variables are linear and normally distributed, and they are also independent of each other. Principal component analysis (PCA) was employed to conduct a rigorous dimensionality reduction analysis.

## 5. Conclusions

The soil NO_3_^−^-N content was uniformly distributed during the reproductive period of chili under ADI conditions, and the uniformity coefficient was higher than that of conventional SDI. ADI significantly reduced the NO_3_^−^-N content in the lower and middle layers of the soil, and there was no significant difference in soil NH_4_^+^-N content among different treatments. ADI significantly promoted root growth and had a positive effect on the yield, plant height, stem diameter, and N accumulation and uptake of chili peppers. In addition, ADI is helpful to improving the quality of chili peppers. In conclusion, ADI can significantly improve the uniformity of soil N distribution, and the appropriate N fertilizer application rate (140 kg hm^−2^) can increase crop nutrient uptake and yield as well as improve the fruit quality of chili peppers.

## Figures and Tables

**Figure 1 plants-13-00642-f001:**
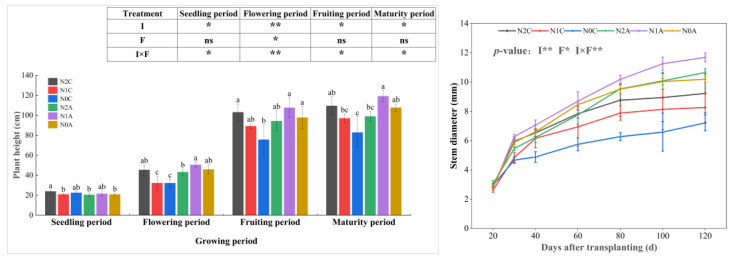
Variations in plant height and stem diameter of chili peppers during the reproductive period. The data represent average values ± standard deviation. N2 represents the normal N application rate, N1 represents the low N application rate, N0 represents no N application, C represents conventional drip irrigation, and A represents ADI. I represents the irrigation method, and F represents the fertilizer amount. Different letters indicate significant differences at the level of *p* < 0.05. * *p* < 0.05. ** *p* < 0.01. ns, not significant.

**Figure 2 plants-13-00642-f002:**
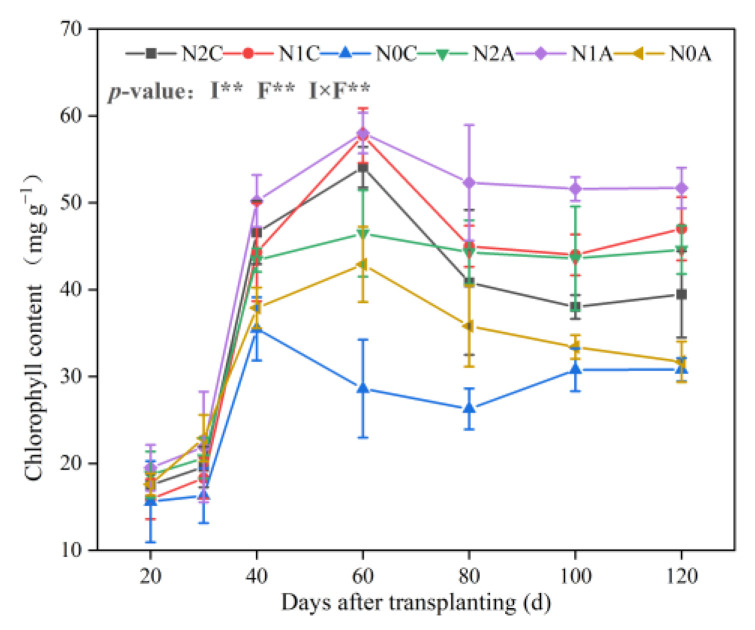
Variations in chlorophyll content of chili pepper leaves. The data represent average values ± standard deviation. N2 represents the normal N application rate, N1 represents the low N application rate, N0 represents no N application, C represents conventional drip irrigation, and A represents ADI. I represents the irrigation method, and F represents the fertilizer amount. Different letters indicate significant differences at the level of *p* < 0.05. ** *p* < 0.01.

**Figure 3 plants-13-00642-f003:**
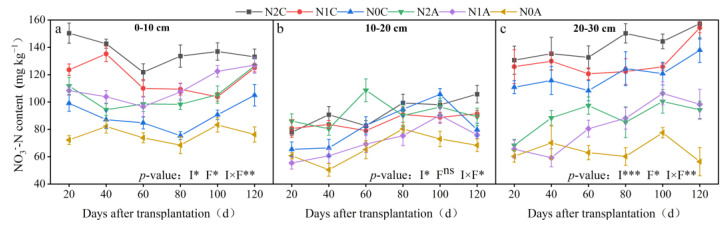
Dynamics of soil NO_3_^−^-N content in each soil layer for different treatments. The data represent average values ± standard deviation. N2 represents the normal N application rate, N1 represents the low N application rate, N0 represents no N application, C represents conventional drip irrigation, and A represents ADI. I represents the irrigation method, and F represents the fertilizer amount. Different letters indicate significant differences at the level of *p* < 0.05. * *p* < 0.05. ** *p* < 0.01. *** *p* < 0.001. ns, not significant. (**a**–**c**) respectively represent the changes in NO_3_^−^−N content within soil layers of 0–10 cm, 10–20 cm, and 20–30 cm.

**Figure 4 plants-13-00642-f004:**
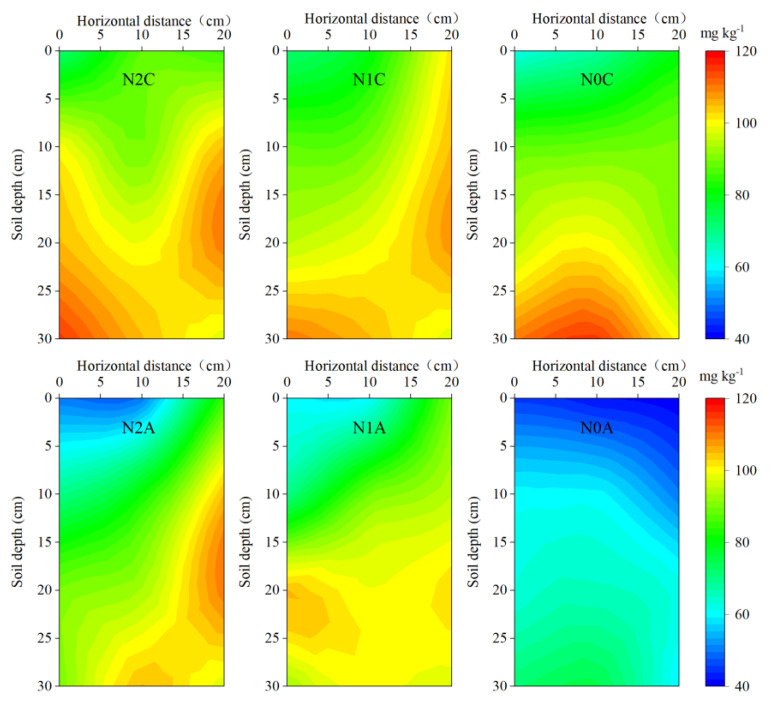
The spatial distribution of soil NO_3_^−^−N content in the fruiting period. N2 represents the normal N application rate, N1 represents the low N application rate, N0 represents no N application, C represents conventional drip irrigation, and A represents ADI. The data depicted in the figure represent the mean values of three replicates for each treatment, with measurements taken at 5 cm intervals horizontally and vertically along the plant roots, resulting in a total of 35 sampling points. The resulting data are presented using interpolation.

**Figure 5 plants-13-00642-f005:**
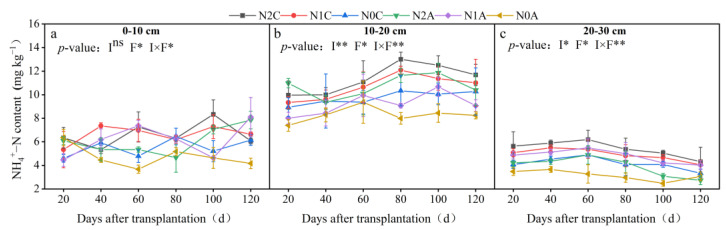
Dynamics of NH_4_^+^−N content in each soil layer for different treatments. The data represent average values ± standard deviation. N2 represents the normal N application rate, N1 represents the low N application rate, N0 represents no N application, C represents conventional drip irrigation, and A represents ADI. I represents the irrigation method, and F represents the fertilizer amount. Different letters indicate significant differences at the level of *p* < 0.05. * *p* < 0.05. ** *p* < 0.01. ns, not significant. (**a**–**c**) respectively represent the changes in NH_4_^+^−N content within soil layers of 0–10 cm, 10–20 cm, and 20–30 cm.

**Figure 6 plants-13-00642-f006:**
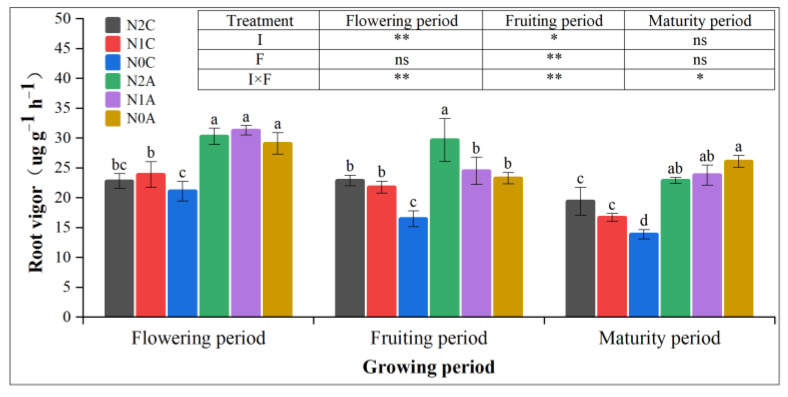
Root vigor in different growth periods of each treatment. The data represent average values ± standard deviation. N2 represents the normal N application rate, N1 represents the low N application rate, N0 represents no N application, C represents conventional drip irrigation, and A represents ADI. I represents the irrigation method, and F represents the fertilizer amount. Different letters indicate significant differences at the level of *p* < 0.05. * *p* < 0.05. ** *p* < 0.01. ns, not significant.

**Figure 7 plants-13-00642-f007:**
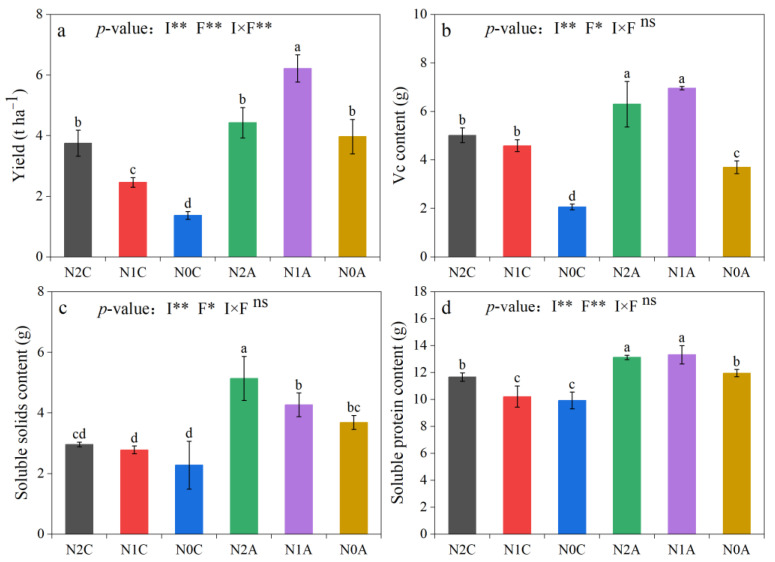
Crop yield (**a**), Vc content (**b**), soluble solids content (**c**), soluble protein content (**d**) in the chili pepper fruit. The data represent average values ± standard deviation. N2 represents the normal N application rate, N1 represents the low N application rate, N0 represents no N application, C represents conventional drip irrigation, and A represents ADI. I represents the irrigation method, and F represents the fertilizer amount. Different letters indicate significant differences at the level of *p* < 0.05. * *p* < 0.05. ** *p* < 0.01. ns, not significant.

**Figure 8 plants-13-00642-f008:**
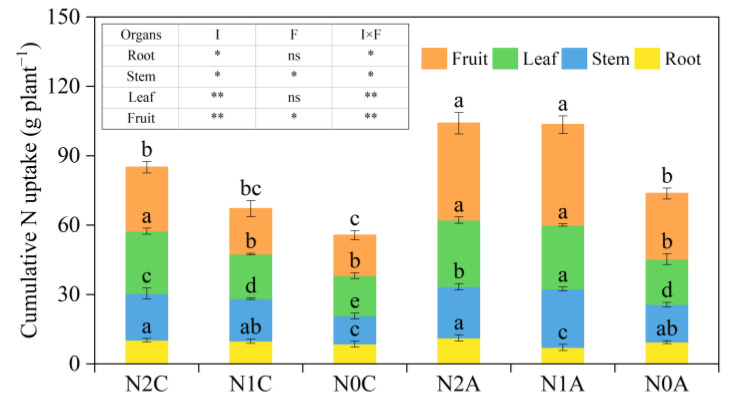
The total N accumulation in the various organs of chili peppers. The data represent average values ± standard deviation. N2 represents the normal N application rate, N1 represents the low N application rate, N0 represents no N application, C represents conventional drip irrigation, and A represents ADI. I represents the irrigation method, and F represents the fertilizer amount. Different letters indicate significant differences at the level of *p* < 0.05. * *p* < 0.05. ** *p* < 0.01. ns, not significant.

**Figure 9 plants-13-00642-f009:**
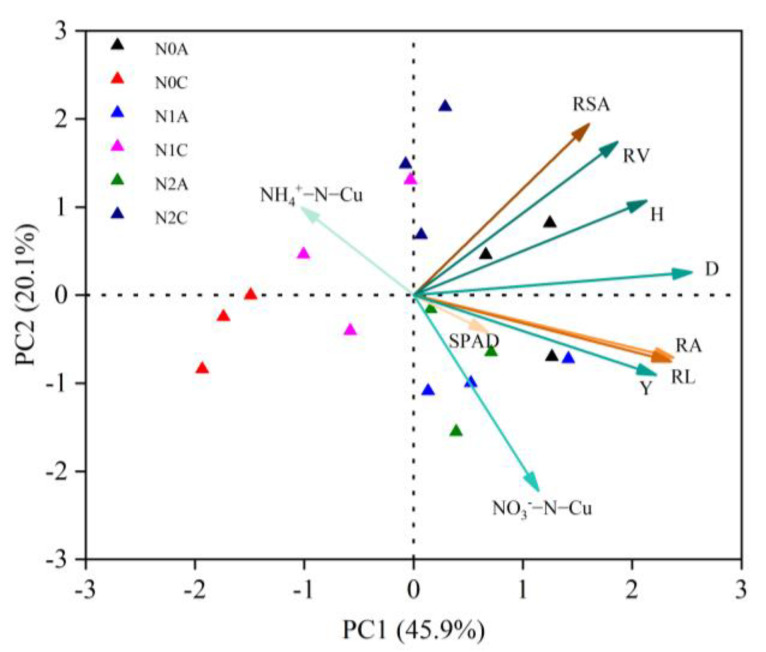
Principal component analysis and variance contribution among indicators. Note: N2 represents the normal N application rate, N1 represents the low N application rate, N0 represents no N application, C represents conventional drip irrigation, and A represents ADI. H, D, NO_3_^−^−N−Cu, NH_4_^+^−N−Cu, SPAD, RA, RL, RSA, RV, and Y stand for plant height, stem diameter, NO_3_^−^−N uniformity coefficient, NH_4_^+^−N uniformity coefficient, chlorophyll content, root vigor, root length, root surface area, root volume, and yield, respectively.

**Figure 10 plants-13-00642-f010:**
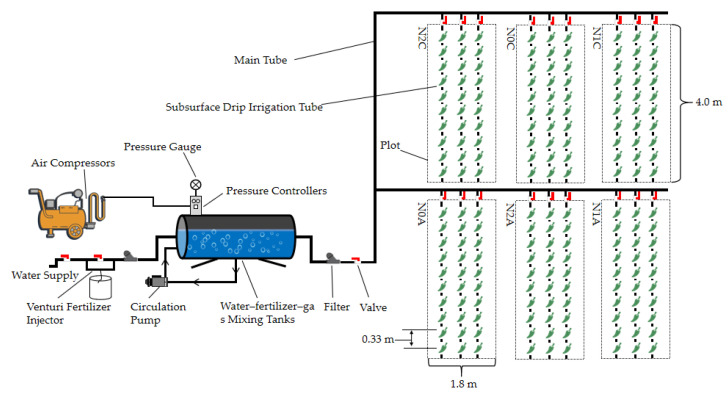
Layout of the greenhouse chili pepper experiment.

**Table 1 plants-13-00642-t001:** Soil NO_3_^−^-N coefficient of distribution uniformity in different treatments in each reproductive period.

Treatment	Seedling Period	Flowering Period	Fruiting Period	Maturity Period
N2C	0.48 ± 0.03 d	0.60 ± 0.03 b	0.62 ± 0.28 c	0.50 ± 0.20 c
N1C	0.67 ± 0.20 c	0.71 ± 0.16 b	0.84 ± 0.10 a	0.75 ± 0.16 b
N0C	0.83 ± 0.18 b	0.76 ± 0.22 b	0.73 ± 0.11 bc	0.70 ± 0.05 bc
N2A	0.86 ± 0.34 b	0.92 ± 0.11 a	0.89 ± 0.18 a	0.95 ± 0.09 a
N1A	0.97 ± 0.07 a	0.92 ± 0.05 a	0.76 ± 0.34 b	0.85 ± 0.15 b
N0A	0.88 ± 0.13 b	0.96 ± 0.14 a	0.86 ± 0.24 a	0.91 ± 0.21 a
I	ns	**	**	ns
F	**	ns	*	**
I × F	*	*	**	*

Note: The data represent average values ± standard deviation. N2 represents the normal N application rate, N1 represents the low N application rate, N0 represents no N application, C represents conventional drip irrigation, and A represents ADI. I represents the irrigation method, and F represents the fertilizer amount. Different letters indicate significant differences at the level of *p* < 0.05. * *p* < 0.05. ** *p* < 0.01. ns, not significant.

**Table 2 plants-13-00642-t002:** Soil NH_4_^+^-N coefficient of distribution uniformity in different treatments in each reproductive period.

Treatment	Seedling Period	Flowering Period	Fruiting Period	Maturity Period
N2C	0.93 ± 0.13 a	0.95 ± 0.11 a	0.82 ± 0.14 b	0.93 ± 0.2 a
N1C	0.94 ± 0.17 a	0.90 ± 0.05 ab	0.93 ± 0.20 a	0.92 ± 0.22 a
N0C	0.88 ± 0.20 b	0.98 ± 0.17 a	0.80 ± 0.24 b	0.96 ± 0.34 a
N2A	0.93 ± 0.14 a	0.89 ± 0.24 b	0.92 ± 0.19 a	0.86 ± 0.14 b
N1A	0.96 ± 0.07 a	0.97 ± 0.18 a	0.95 ± 0.06 a	0.92 ± 0.07 b
N0A	0.92 ± 0.30 a	0.90 ± 0.05 a	0.94 ± 0.19 a	0.96 ± 0.08 a
I	*	**	*	ns
F	ns	*	ns	*
I × F	*	*	*	**

Note: The data represent average values ± standard deviation. N2 represents the normal N application rate, N1 represents the low N application rate, N0 represents no N application, C represents conventional drip irrigation, and A represents ADI. I represents the irrigation method, and F represents the fertilizer amount. Different letters indicate significant differences at the level of *p* < 0.05. * *p* < 0.05. ** *p* < 0.01. ns, not significant.

**Table 3 plants-13-00642-t003:** Indicators of the root system of chili peppers in different treatments.

Treatment	Root Length (cm)	Root Surface Area (cm^2^)	Root Volume (cm^3^)
N0C	392.54 ± 56.35 c	207.37 ± 45.27 c	25.14 ± 3.68 c
N1C	509.34 ± 49.38 b	256.34 ± 65.2 bc	26.59 ± 9.30 c
N2C	625.18 ± 168.36 cb	342.58 ± 69.34 ab	32.45 ± 9.64 bc
N0A	660.28 ± 64.25 b	286.19 ± 46.56 bc	38.50 ± 8.16 b
N1A	906.46 ± 75.60 a	491.35 ± 48.33 a	53.32 ± 6.48 ab
N2A	788.36 ± 83.4 ab	446.30 ± 24.68 a	44.59 ± 5.36 ab
I	**	ns	ns
F	**	ns	*
I×F	**	**	**

Note: The data represent average values ± standard deviation. N2 represents the normal N application rate, N1 represents the low N application rate, N0 represents no N application, C represents conventional drip irrigation, and A represents ADI. I represents the irrigation method, and F represents the fertilizer amount. Different letters indicate significant differences at the level of *p* < 0.05. * *p* < 0.05. ** *p* < 0.01. ns, not significant.

**Table 4 plants-13-00642-t004:** Experimental design.

Treatment	N Fertilizer Usage (kg hm^−2^)	Base Fertilizer	N Topdressing Dosage Ratio	Ventilation Ratio
N0A	0	150-P_2_O_5_ 200-K_2_O	2:3:2:3:1	15%
N1A	140			15%
N2A	210			15%
N0C	0			0
N1C	140			0
N2C	210			0

## Data Availability

Data are contained within the article.

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
