# Peer review of "Effects of Aerated Drip Irrigation on the Soil Nitrogen Distribution, Crop Growth, and Yield of Chili Peppers"

_plants, 2024, doi:10.3390/plants13050642_

Round 1
Reviewer 1 Report
Comments and Suggestions for Authors
The manuscript of Lei et al. deals with the effect of aerated drip irrigation on soil nitrogen distribution and chili pepper nitrogen uptake applying different nitrogen fertilization regimes.
The analyses carried out by the authors are overall complete.
However, the manuscript needs a deep revision. The manuscript is in general difficult to read, too long sentences, problems with English grammar, statistic included in the text rather than in the figures, wrong table numbering, different colors in every figure, etc. I reported several suggestions in the attached file. Please carefully revise the manuscript accordingly. I recommend including the statistic in the figures. Two way ANOVA P-values are always missing thus it is impossibile to verify if the post-hoc lettering is accurate and if the effect of the interaction between irrigation type and N fertilization or only the irrigation type is significant. Thus, it is really difficult to verify if conclusions are supported by results. I also recommend to reduce the sentence lengths and overall try to shorten the text.

Comments on the Quality of English Language
English language must be revised.
Reviewer 2 Report
Comments and Suggestions for Authors
Thank you for entrusting me with a review of this work. It is an article containing interesting and new information written in good language substantively and scientifically. The literature is well chosen, the introduction is extensive and sufficient. However, I ask the authors for some clarification. What was the reason for the use of nitrogen as urea? Why was this fertilizer chosen? if an EC and pH analysis of the soil was done, it would be useful. In Table 1, the ratio of nitrogen and phosphorus and potassium fertilization is inlegibly markedit looks like it only applies to N2A and N0C. Variation of NO3--N content in each soil layer of different treatments is better presented in tables. It is also wrong to write the ionic form and then add the letter N. Please refer to these comments
Reviewer 3 Report
Comments and Suggestions for Authors
The article is well written. Interesting. English should be improved, use synonyms to avoid repetition of words.
Comments on the Quality of English Language
The article is well written. Interesting. English should be improved, use synonyms to avoid repetition of words.
Author Response
Thank you for reviewing our manuscript, the manuscript was revised based on your comments.
Reviewer 4 Report
Comments and Suggestions for Authors
The paper is very interesting. Well articulated and the methods used are appropriate. It moves in the context of both soil chemistry and agricultural hydraulics and, although the experiment was conducted in a greenhouse, it opens up interesting scenarios for open field application. The proposed irrigation system is very interesting because it is related to the loss of nitrogen in the form of nitrate, which constitutes a further topic of discussion and concern. In my opinion the paper can be accepted in this form
Author Response

(The authors gave the same response as above.)

Round 2
Reviewer 1 Report
Comments and Suggestions for Authors
Dear authors,
the manuscript is improved from its first version. However, I reported several suggestions in the attached file to further improve it.
Major comments, in addition to those in attachment:
I believe an extensive editing of English language is still required.
A complete and clear statistical analysis is still lacking in several figures.
How Figure 5 was realized must be added to the manuscript (more details are in the attached file)
Discussion must be improved comparing results with literature. In some parts it is just a repetition of results.
Minor comments: abstract is still too long for the journal guidelines, the organization of figures is still too confusing and the figures are graphically too different each others although they are improved compare to the previous manuscript version. Please improve figure and table captions. I have also still not clear how the treatments are organized in the experimental area, please indicate the distribution of replicates in Figure 10. Abbreviations must be used consistently throughout the manuscript.

Comments on the Quality of English Language
Author Response
Dear reviewers:
Thank you for taking time out of your busy schedule to review our manuscript. We have revised it based on your suggestions and highlighted the changes in the new version, which you can find in the attached file. Furthermore, as we are celebrating the Chinese New Year these days, I would like to extend my best wishes to you for a happy holiday and continued success!
Sincerely

Round 3
Reviewer 1 Report
Comments and Suggestions for Authors
The manuscript is much improved from the previous version.
There are still some things to be revised, I have provided some comments in the attached file. The statistics applied in Figure 1a must be applied also in Figure 1b, 2 , and 5. Please remove F-values from the figures, the P-values are enough. Please use the abbreviations for N, N-NO3 and N-NH4 consistently throughout the manuscript.
For future experiments, please consider that replicates must be randomly distributed in the experimental area. The three replicates cannot be next to each other as reported in Figure 10.

Comments on the Quality of English Language
An english editing by a professional service is still needed.
Author Response

(The authors gave the same response as above.)
